# Adversarially Perturbed Batch Normalization: A Simple Way to Improve Image Recognition

## Abstract

Recently, it has been shown that adversarial training (AT) by injecting adversarial samples can improve the quality of recognition. However, the existing AT methods suffer from the performance degradation on the benign samples, leading to a gap between robustness and generalization. We argue that this gap is caused by the inaccurate estimation of the Batch Normalization (BN) layer, due to the distributional discrepancy between the training and test set. To bridge this gap, this paper identifies the adversarial robustness against the indispensable noise in BN statistics. In particular, we proposed a novel strategy that adversarially perturbs the BN layer, termed ARAPT. The ARAPT leverages the gradients to shift BN statistics and helps models resist the shifted statistics to enhance robustness to noise. Then, we introduce ARAPT into a new paradigm of AT called model-based AT, which strengthens models' tolerance to noise in BN. Experiments indicate that the APART can improve model generalization, leading to significant improvements in accuracy on benchmarks like CIFAR-10, CIFAR-100, Tiny-ImageNet, and ImageNet.

## 1 Introduction

Recent works [1, 2, 3, 4] show that deep neural networks are sensitive to adversarial perturbations, which gives rise to the rapid development of adversarial training (AT) methods [5, 6, 7, 8]. These AT methods enhance models' robustness against adversarial samples by solving a min-max optimization problem [5]. However, many efforts [6, 9, 10] have corroborated that there is a trade-off between standard and robust accuracy, in which AT usually degrades models' performance on benign samples, though they enjoy the accuracy gains on adversarial samples.

Xie *et al*. [11] challenges the widely accepted idea that AT hurts models' generalization. They proposed adversarial propagation (AdvProp) to exploit the adversarial features via auxiliary Batch Normalization (BN) [12] layers. However, the huge computational overhead discourages more efforts in its applications. Thus, the further work proposed fast AdvProp [13] to reduce the computations by leveraging the acceleration of AT [14]. Nevertheless, leveraging adversarial samples to perform AT in *non-safety* tasks leads to the following question:

*What does the model defend against?*

Indeed, adversarially trained models appear closely tied to the robustness against adversarial attacks for safety concerns. However, in non-safety situations, there is an open question of what these models defend against. This question is essentially related to the gap between models' generalization and adversarial robustness. Prior efforts [15, 16, 17] reposition the adversarial robustness as the

robustness against the worst-case unseen domains, in attempts to bridge the generalization-robustness gap. They usually enhance models' robustness to adversarially generated domains to improve the generalization. Nonetheless, there is inevitably a mismatch between such generated domains and the actual domains. The mismatch hinders their further applications.

In this paper, we answer this question by identifying the robustness against the noise of BN statistics that are the estimated mean and variance. The statistics noise is indispensable due to the distributional discrepancy between training and test domains [18, 19, 20, 21, 22, 23]. Moreover, insufficient batch size will cause the severer noise in some computation-demanding tasks [18, 19, 20]. In this study, we cast the statistics noise as a numerical problem to avoid the issue of how to match the adversarially generated domains with the actual ones. Since the noise degrades BN's performance, numerically strengthening models' tolerance to such noise will boost the generalization of BN-based models.

In this work, we train models by **A**dversarially **P**erturbed b**A**tch no**R**maliza**T**ion (APART) that perturbs BN statistics and updates the model parameters to resist the perturbation on the fly. More concretely, APART performs backward passes twice over each batch of benign samples. The first backward pass produces two gradient computations: one is normal gradient that helps update parameters of model *w.r.t.* samples' patterns, and the other one is statistics gradient that is used to perturb the statistics parameters in BN. Then, the second pass is performed to generate the defensive gradient that helps the model resist the adversarial statistics perturbation. The normal and defensive gradients are combined to improve both generalization and robustness of the model. All gradients are computed by the regular gradient descent algorithm. Note that APART combines the normal gradient with the defensive one without changing the update strategy and without crafting the adversarial samples. This process follows a paradigm of AT performing attacks and defense within models instead of on samples, hence the name model-based AT. Besides, as suggested by AdvProp [11], the BN statistics computed over the adversarial passes are dropped to avoid the corruption.

Experimentally, APART makes models less brittle to noisy BN statistics. As a consequence, the models enjoy significant accuracy gains on CIFAR [24], Tiny-ImageNet [25] and ImageNet [26] datasets. Moreover, the improvement brought by APART only depends on BN, allowing the combination with other training methods, *e.g.* data augmentation [27] and sharpness-aware minimization [28].

**Summary of contributions**:

- We identify the adversarial robustness against the noise in BN statistics to bridge the gap between models' generalization and robustness. Enhancing such robustness by AT improves models' generalization on benign samples.

- We proposed APART to achieve the robustness against the statistics noise. APART follows a new paradigm of AT utilizing the gradients efficiently. By strengthening BN-based models' tolerance to BN statistics noise, APART significantly improves the models' performance.

- With its plug-and-play nature, APART allows the combination with other training methods and enjoys the further accuracy gains.

## 2  Related Work

### 2.1  Adversarial Training

Adversarial training (AT) [1, 2, 5, 29] is empirically demonstrated to be one of the most effective defense methods for models' safety concerns. Instead, many non-AT methods [30, 31, 32, 33] fail to defend against adaptive attacks [4]. However, AT sacrifices the standard accuracy on benign samples to increase models' robustness [9]. Thus, there is a trade-off between the robustness and generalization [6]. Furthermore, many efforts [34, 9, 35, 36, 37] theoretically and experimentally corroborate the difficulty of achieving adversarial robustness over limited data. Besides adversarial robustness, other works [38, 14] focus on the efficiency of AT due to the high computational overhead of vanilla AT methods [5, 6]. The proposed fast AT method [14] accelerates the training in a

simple way, but suffers from catastrophic overfitting [39, 40]. This problem gives rise to more efforts [39, 40, 41, 42].

Besides performing AT over samples, Adversarial Weight Perturbation [8] additionally perturbs parameters to enhance the generalization from a perspective of loss landscape. In non-safety tasks, some efforts [28, 43, 44, 45] have been devoted to such parameter-based AT and show promise in improving models' generalization. In this study, the proposed method follows a more generic paradigm of AT that allows the attacks on each component of models even including the non-parameter BN statistics.

## 2.2 Adversarial Robustness Beyond Safety

Though the disadvantage of AT in models' generalization discourages the efforts of its non-safety applications, Xie *et al.* [11] proposed AdvProp to challenge this issue. AdvProp utilizes auxiliary BN layers to avoid corrupting the BN statistics estimated over benign samples. In this manner, AdvProp improves models' generalization and inspires the further studies [46, 16, 47, 15, 48, 49] of the adversarial robustness beyond safety. Indeed, AT provides the framework of crafting and countering the worse-case unseen domains [15, 16, 17], and enhances adversarial robustness varying in different contexts. Besides, Mei *et al.* [13] utilize the acceleration of fast AT [14] to significantly reduce the computational overhead of AdvProp [11].

In this work, the proposed AT method, termed APART, increases models' robustness against the noise of BN statistics. Though the perturbation formula of the statistics is somewhat similar to that of Adversarial Batch Normalization (AdvBN) [15], there are three major differences between them in implementations: 1) APART perturbs the entire network by slightly changing each BN layer, instead of perturbing the features generated from a specific non-BN layer [15]; 2) in each iteration, APART performs backward passes only twice to carry out the attack and defense efficiently, instead of performing multiple backward passes inefficiently [15]; 3) APART trains each model from the scratch, instead of fine-tuning a pre-trained model [15], which leads to incomparability between APART and AdvBN.

## 2.3 Normalization

Batch Normalization (BN) [12] has successfully boosted a broad range of deep neural networks by accelerating the training. However, the noisy statistics of BN degrade its performance experimentally [20] and theoretically [50]. Many efforts have been devoted to more accurate estimators of the statistics [18, 19, 20, 21, 22, 23]. Some estimators perform the normalization along different axes, *e.g.* Layer Normalization [18], Instance Normalization [19] and Group Normalization [20]. They reduce the noise in the case of tiny batch but suffer from performance degradation under large batch as the alternative to BN. More efforts [21, 51, 52, 16] exploit the combination of these normalization methods. They selectively use the axis-specific statistics to perform normalization in response to different domains. Additionally, the on-the-fly estimation of BN statistics over adversarial samples is experimentally found to have negative impacts on the standard accuracy [11, 53]. This finding leads to more exploration in BN under AT [11, 13, 15, 46, 49].

The noisy statistics result from a mismatch between the seen and unseen domains and are therefore indispensable without the domain-specific knowledge. Furthermore, tiny batch size caused by the computation-demanding tasks results in the severer noise. From an opposite perspective of these methods denoising the statistics, our method hardens BN-based models' robustness against the noise.

## 3 Method

In this section, we firstly introduce a new paradigm of AT that allows us to perform attacks and defense within models rather than on samples. Then, we propose APART to implement this paradigm in a simple way. Finally, we discuss the enhancement of APART, which is derived from the potential link between APART and the other training method.

## 3.1 Model-Based Adversarial Training

The vanilla AT formulates a min-max game [5] by adversarially crafting and defensively countering the imperceptible perturbations to samples. Specifically, given the ground truth $\boldsymbol{y}$ and sample $\boldsymbol{x}$'s allowed neighborhood $\mathcal{S}(\boldsymbol{x})$, we minimize the expectation of a $\boldsymbol{\theta}$-parameterized loss $\mathcal{L}(\boldsymbol{x}^*, \boldsymbol{y}; \boldsymbol{\theta})$ with $\boldsymbol{x}^* \in \mathcal{S}(\boldsymbol{x})$ maximizing $\mathcal{L}(\cdot, \boldsymbol{y}; \boldsymbol{\theta})$, *i.e.*,

$$\min_{\boldsymbol{\theta}} \mathbb{E}_{\boldsymbol{x},\boldsymbol{y}} \mathcal{L}(\boldsymbol{x}^*, \boldsymbol{y}; \boldsymbol{\theta}), \quad \text{where} \quad \boldsymbol{x}^* := \operatorname*{argmax}_{\boldsymbol{x}' \in \mathcal{S}(\boldsymbol{x})} \mathcal{L}(\boldsymbol{x}', \boldsymbol{y}; \boldsymbol{\theta}). \tag{1}$$

Empirically, the maximization of Eq. (1) is achieved by a gradient ascent method for each sample. The gradient $\nabla_{\boldsymbol{x}} \mathcal{L}(\boldsymbol{x}, \boldsymbol{y}; \boldsymbol{\theta})$ is iteratively computed by full forward and backward passes on the model. This process merely requires the inputs' gradients $\nabla_{\boldsymbol{x}} \mathcal{L}(\boldsymbol{x}, \boldsymbol{y}; \boldsymbol{\theta})$ and drops all the internal gradients $\nabla_{\boldsymbol{\theta}} \mathcal{L}(\boldsymbol{x}, \boldsymbol{y}; \boldsymbol{\theta})$ without their further utilization after finishing the backward pass. Thus, such vanilla AT suffers from low efficiency of utilizing the gradients. Meanwhile, AT's potency is limited by such sample-based attacks and defense. Therefore, we propose a paradigm of *model-based* AT to leverage internal gradients efficiently and allow the attacks and defense within models.

To perform such AT, each component of a model is categorized into two types: one for the attacker, and one for the defender. Denoting by $\boldsymbol{\theta}, \boldsymbol{\phi}$ the parameters of the adversarial and defensive components respectively, we formulate the model-based AT as follows

$$\min_{\boldsymbol{\phi}} \mathbb{E}_{(\boldsymbol{x},\mathbf{y})} \left[ \mathcal{R}(\boldsymbol{x}, \boldsymbol{y}; \boldsymbol{\phi}) + \max_{\boldsymbol{\theta} \in \boldsymbol{\Theta}} \mathcal{L}(\boldsymbol{x}, \boldsymbol{y}; \boldsymbol{\theta}, \boldsymbol{\phi}) \right], \tag{2}$$

where $\boldsymbol{\Theta}$ is a parameter space that can be bounded to avoid trivial results; $\mathcal{R}(\boldsymbol{x}, \boldsymbol{y}; \boldsymbol{\phi})$ is a task-specific loss allowing models to learn the normal patterns in samples, and $\mathcal{L}(\boldsymbol{x}, \boldsymbol{y}; \boldsymbol{\theta}, \boldsymbol{\phi})$ can share a similar formulation of $\mathcal{R}$ to enable more pattern exploration in an adversarial manner. Overall, Eq. (2) provides a generic formulation of model-based AT. For example, Generative Adversarial Networks (GANs) [54] can be repositioned as a special case of such AT, in which the generator and discriminator are regarded as the attacker and defender respectively, and the discriminative losses are cast as proper $\mathcal{R}$ and $\mathcal{L}$ in Eq. (2). Next, we introduce APART to implement this model-based AT in a simple way.

## 3.2 Adversarially Perturbed Batch Normalization

Model-based AT helps models harden their robustness against a specific problem. We shift our attention to the noisy BN statistics [20], and apply the proposed AT to address this problem.

Firstly, we embed two temporary parameters $\boldsymbol{\delta}_{\boldsymbol{\mu}}, \boldsymbol{\delta}_{\boldsymbol{\sigma}}$ into each BN layer as the adversarial parameters $\boldsymbol{\theta}$ in Eq. (2), which will be dropped after the training. Inspired by AdvBN [15], with $\boldsymbol{\delta}_{\boldsymbol{\mu}}, \boldsymbol{\delta}_{\boldsymbol{\sigma}} \leftarrow \mathbf{0}$, we reformulate the BN mapping as

$$\mathbf{BN}(\boldsymbol{x}; \boldsymbol{\delta}_{\boldsymbol{\mu}}, \boldsymbol{\delta}_{\boldsymbol{\sigma}}) = \boldsymbol{\gamma}(\mathbf{1} + \boldsymbol{\delta}_{\boldsymbol{\sigma}}) \cdot \frac{\boldsymbol{x} - (\mathbf{1} + \boldsymbol{\delta}_{\boldsymbol{\mu}})\hat{\boldsymbol{\mu}}}{\hat{\boldsymbol{\sigma}}} + \boldsymbol{\beta}, \tag{3}$$

where each operator is element-wise; $\hat{\boldsymbol{\mu}}, \hat{\boldsymbol{\sigma}}$ are the mean and standard deviation estimated over a batch of $\boldsymbol{x}$ respectively; $\boldsymbol{\gamma}, \boldsymbol{\beta}$ are the parameters of BN's affine mapping. We bound the $d$-dimensional $\boldsymbol{\delta}_{\boldsymbol{\mu}}, \boldsymbol{\delta}_{\boldsymbol{\sigma}}$ such that $\boldsymbol{\delta}_{\boldsymbol{\mu}}, \boldsymbol{\delta}_{\boldsymbol{\sigma}} \in [-\epsilon, \epsilon]^d$ for a small sufficient perturbation radius $\epsilon > 0$. The bound avoids trivial results, *e.g.* $\mathbf{BN}(\boldsymbol{x}; \boldsymbol{\delta}_{\boldsymbol{\mu}}, \boldsymbol{\delta}_{\boldsymbol{\sigma}})|_{\boldsymbol{\delta}_{\boldsymbol{\sigma}}=-\mathbf{1}} \equiv \mathbf{0}$. Now all the other trainable parameters within the entire model including $\boldsymbol{\gamma}, \boldsymbol{\beta}$ are naturally the defensive parameters $\boldsymbol{\phi}$. The losses $\mathcal{R}, \mathcal{L}$ in Eq. (2) are both the same task-specific loss, *i.e.*, the cross entropy in recognition. Therefore, Eq. (3) implies ($\boldsymbol{\theta}$ indicates all BN layers' $\boldsymbol{\delta}_{\boldsymbol{\mu}}, \boldsymbol{\delta}_{\boldsymbol{\sigma}}$)

$$\mathcal{L}(\boldsymbol{x}, \boldsymbol{y}; \boldsymbol{\theta}, \boldsymbol{\phi})|_{\boldsymbol{\theta}=\mathbf{0}} = \mathcal{R}(\boldsymbol{x}, \boldsymbol{y}; \boldsymbol{\phi}) \quad \nabla_{\boldsymbol{\phi}} \mathcal{L}(\boldsymbol{x}, \boldsymbol{y}; \boldsymbol{\theta}, \boldsymbol{\phi})|_{\boldsymbol{\theta}=\mathbf{0}} = \nabla_{\boldsymbol{\phi}} \mathcal{R}(\boldsymbol{x}, \boldsymbol{y}; \boldsymbol{\phi}), \tag{4}$$

by which we can get both $\nabla_{\boldsymbol{\theta}} \mathcal{L}(\boldsymbol{x}, \boldsymbol{y}; \boldsymbol{\theta}, \boldsymbol{\phi})|_{\boldsymbol{\theta}=\mathbf{0}}$ and $\nabla_{\boldsymbol{\phi}} \mathcal{R}(\boldsymbol{x}, \boldsymbol{y}; \boldsymbol{\phi})$ in a single backward pass with $\mathcal{L}(\boldsymbol{x}, \boldsymbol{y}; \boldsymbol{\theta}, \boldsymbol{\phi})|_{\boldsymbol{\theta}=\mathbf{0}}$ as the loss.

Secondly, we propose APART that follows a gradient accumulation strategy, instead of the alternative update strategy used by GANs [54]. Specifically, in each iteration of a normal gradient descent algorithm over a batch of samples $\mathcal{D} := \{(\boldsymbol{x}_i, \boldsymbol{y}_i), 1 \le i \le M\}$,

**Algorithm 1:** Pseudo code of APART getting the gradient for a batch of samples, given some perturbation radius $\epsilon$, number of samples in the second pass $N$, and group number $n$

**Data:** A batch of samples $\mathcal{D} := \{(\boldsymbol{x}_i, \boldsymbol{y}_i), 1 \leq i \leq M\}$
**Result:** The gradient $\boldsymbol{g}$ for this batch of samples

1   $\boldsymbol{\theta} \leftarrow \boldsymbol{0}$
2   Perform forward and backward passes once over $\mathcal{D}$, generating $\boldsymbol{g_\theta}$ and $\boldsymbol{g_\phi}$ simultaneously
3      $\boldsymbol{g_\theta} \leftarrow \mathbb{E}_{(\boldsymbol{x}_i, \boldsymbol{y}_i) \in \mathcal{D}} \nabla_\theta \mathcal{L}(\boldsymbol{x}_i, \boldsymbol{y}_i; \boldsymbol{\theta}, \boldsymbol{\phi})|_{\boldsymbol{\theta}=\boldsymbol{0}}$
4      $\boldsymbol{g_\phi} \leftarrow \mathbb{E}_{(\boldsymbol{x}_i, \boldsymbol{y}_i) \in \mathcal{D}} \nabla_\phi \mathcal{R}(\boldsymbol{x}_i, \boldsymbol{y}_i; \boldsymbol{\phi})$
5   $\boldsymbol{\theta} \leftarrow \epsilon \mathrm{sign}(\boldsymbol{g_\theta})$
6   Randomly draw $N$ samples $\mathcal{S} \subseteq \mathcal{D}$
7   Group $\mathcal{S}$ into $n$ equally sized subsets $\mathcal{S}_1, \mathcal{S}_2, \ldots, \mathcal{S}_n$
8   $\boldsymbol{h_\phi} \leftarrow \boldsymbol{0}$
9   **for** $j \leftarrow 1$ **to** $n$ **do**
10    |   $\boldsymbol{h_\phi} \leftarrow \boldsymbol{h_\phi} + \frac{1}{n} \mathbb{E}_{(\boldsymbol{x}_i, \boldsymbol{y}_i) \in \mathcal{S}_j} \nabla_\phi \mathcal{L}(\boldsymbol{x}_i, \boldsymbol{y}_i; \boldsymbol{\theta}, \boldsymbol{\phi})|_{\boldsymbol{\theta}=\epsilon \mathrm{sign}(\boldsymbol{g_\theta})}$
11   **end**
12   $\boldsymbol{g} \leftarrow \frac{M}{M+N} \boldsymbol{g_\phi} + \frac{N}{M+N} \boldsymbol{h_\phi}$

- **Step 1:** With $\boldsymbol{\theta} \leftarrow \boldsymbol{0}$, APART performs the forward and backward passes once over this batch of samples to generate the gradients *w.r.t.* the adversarial and defensive parameters, *i.e.*, $\boldsymbol{g_\theta} := \mathbb{E}_{(\boldsymbol{x}_i, \boldsymbol{y}_i) \in \mathcal{D}} \nabla_\theta \mathcal{L}(\boldsymbol{x}_i, \boldsymbol{y}_i; \boldsymbol{\theta}, \boldsymbol{\phi})|_{\boldsymbol{\theta}=\boldsymbol{0}}$ and $\boldsymbol{g_\phi} := \mathbb{E}_{(\boldsymbol{x}_i, \boldsymbol{y}_i) \in \mathcal{D}} \nabla_\phi \mathcal{R}(\boldsymbol{x}_i, \boldsymbol{y}_i; \boldsymbol{\phi})$ according to Eq. (4). Like Fast Gradient Sign Method [2], APART uses $\epsilon \mathrm{sign}(\boldsymbol{g_\theta})$ to assign $\boldsymbol{\theta}$, which empirically performs the inner maximization of Eq. (2) and generates the adversarially perturbed statistics in each BN layer.

- **Step 2:** With the adversarial BN statistics, APART performs the forward and backward passes again, over a full/incomplete batch of the same samples $\mathcal{S} \subseteq \mathcal{D}$. This backward pass yields the gradient resisting the attack, *i.e.*, $\boldsymbol{h_\phi} := \mathbb{E}_{(\boldsymbol{x}_i, \boldsymbol{y}_i) \in \mathcal{S}} \nabla_\phi \mathcal{L}(\boldsymbol{x}_i, \boldsymbol{y}_i; \boldsymbol{\theta}, \boldsymbol{\phi})|_{\boldsymbol{\theta}=\epsilon \mathrm{sign}(\boldsymbol{g_\theta})}$. The weighted gradient $\boldsymbol{g} := (1-r)\boldsymbol{g_\phi} + r\boldsymbol{h_\phi}$ is finally used in the outer minimization of Eq. (2) for this batch of samples, where $r \in [0, 0.5]$ re-balances the gradients.

Apparently, using a full batch of the samples in the second pass leads to the best performance, but results in more computational overhead. Instead, using the incomplete batch of these samples allows the less computation but suffers from insufficient defense against the attack. Thus, the ratio $r = N/(M+N)$ is introduced to re-balance the gradients with $N = |\mathcal{S}|$ the number of the samples used in the second pass. Additionally, the on-the-fly BN statistics estimated in the second pass are completely dropped to avoid corrupting the statistics at inference, like the auxiliary BN layers [11].

Note that stronger attacks in AT indirectly enhance the adversarial robustness [5]. Thus, we slightly modify the process of the second pass to strengthen the attack. The modification increases the noise in the adversarial BN statistics without additional computation. In details, we group the samples into equally sized sets and stop their group-to-group communications in BN layers during the second forward pass. This is inspired by the fact that smaller batch size results in larger noise in the statistics. In this manner, the BN statistics are estimated over less samples without reducing the entire batch size, giving rise to the less adversarial accuracy.

Overall, APART only changes the way of getting gradients in each iteration, without involving in data augmentation or network modification. Therefore, APART has plug-and-play nature that allows the combination with a broad range of training methods. We summarize APART in Algorithm 1, and then introduce the enhancement of APART.

### 3.3   Enhancement by Combination with Sharpness-Aware Minimization

Besides APART, Sharpness-Aware Minimization (SAM) [28] also belongs to and use the proposed model-based AT paradigm. SAM improves network training from a perspective of loss landscape relating to models' generalization. Given the empirical loss function $\mathcal{L}_\mathcal{D}$ estimated over a dataset $\mathcal{D}$,

SAM minimizes $\mathcal{L}_\mathcal{D}$ with a sharpness measure:

$$\min_{\boldsymbol{w}} \left[ \max_{||\boldsymbol{\delta}||_2 \leq \rho} \mathcal{L}_\mathcal{D}(\boldsymbol{w} + \boldsymbol{\delta}) - \mathcal{L}_\mathcal{D}(\boldsymbol{w}) \right] + \mathcal{L}_\mathcal{D}(\boldsymbol{w}) + \frac{\lambda}{2}||\boldsymbol{w}||_2^2, \tag{5}$$

where $\boldsymbol{w}$ is the model's trainable parameters; $\rho > 0$ is small sufficient to restrict the perturbation $\boldsymbol{\delta}$; $\lambda > 0$ is used to control the regularizer $||\boldsymbol{w}||_2^2$; the term in the square bracket measures $\mathcal{L}_\mathcal{D}$'s sharpness. Obviously, Eq. (5) is equivalent to

$$\min_{\boldsymbol{w}} \max_{||\boldsymbol{\delta}||_2 \leq \rho} \mathcal{L}_\mathcal{D}(\boldsymbol{w} + \boldsymbol{\delta}) + \frac{\lambda}{2}||\boldsymbol{w}||_2^2. \tag{6}$$

Actually, Eq.(6) performs the model-based AT formulated by Eq. (2), in which we treat $\boldsymbol{\delta}, \boldsymbol{w}$ as the adversarial and defensive parameters $\boldsymbol{\theta}, \boldsymbol{\phi}$ respectively, and let $\mathcal{L}(\boldsymbol{x}, \boldsymbol{y}; \boldsymbol{\theta}, \boldsymbol{\phi}) = \mathcal{L}_\mathcal{D}(\boldsymbol{w} + \boldsymbol{\delta}), \mathcal{R}(\boldsymbol{x}, \boldsymbol{y}; \boldsymbol{\phi}) \equiv 0$ with the regularizer $\frac{\lambda}{2}||\boldsymbol{w}||_2^2$ included in the empirical optimization. Therefore, SAM and APART essentially share the same training paradigm. Surprisingly, these two methods implement this paradigm in a complementary way: SAM focuses on the trainable parameters optimized by gradient descent, while APART concentrates on the non-trainable BN statistics requiring estimation instead of optimization. The training paradigm enables them to enhance models' robustness in different contexts, which inspires a combination of them.

Note that the inner maximization in Eq. (6) is approximately achieved by one-step gradient ascent. Then, the outer minimization is performed by estimating the gradient *w.r.t.* the adversarially shifted parameters $\boldsymbol{w} \leftarrow \boldsymbol{w} + \boldsymbol{\delta}$ [28]. Thus, SAM shares a similar two-step strategy of APART. Such similarity allows us to perform APART and SAM simultaneously by a slight modification of Algorithm 1, termed APART-SAM. Specifically, we adopt the weights' perturbations, *i.e.*, Eq. (2) therein [28], reformulated as

$$\hat{\boldsymbol{\delta}}(\boldsymbol{\phi}) = \rho \, \text{sign}\left(\nabla_{\boldsymbol{\phi}} \mathcal{R}(\boldsymbol{x}, \boldsymbol{y}; \boldsymbol{\phi})\right) |\nabla_{\boldsymbol{\phi}} \mathcal{R}(\boldsymbol{x}, \boldsymbol{y}; \boldsymbol{\phi})|^{q-1} / \left( ||\nabla_{\boldsymbol{\phi}} \mathcal{R}(\boldsymbol{x}, \boldsymbol{y}; \boldsymbol{\phi})||_q^q \right)^{1/p}, \tag{7}$$

where $1/p + 1/q = 1$ and experimentally let $p = 2$ as suggested by [28]. Then, we modify APART's first step by additionally perturbing the defensive parameters $\boldsymbol{\phi} \leftarrow \boldsymbol{\phi} + \hat{\boldsymbol{\delta}}(\boldsymbol{\phi})$ to enhance the attacks with the second step unchanged. The additional perturbation $\hat{\boldsymbol{\delta}}(\boldsymbol{\phi})$ just employs the gradient $\nabla_{\boldsymbol{\phi}} \mathcal{R}(\boldsymbol{x}, \boldsymbol{y}; \boldsymbol{\phi})$ previously computed by APART's first step, and normalizes them with ignorable extra computations. Therefore, APART-SAM is computation-friendly enhancement of APART.

## 4 Experiments

### 4.1 Experimental Setup

**Datasets and Models.** We evaluate APART on CIFAR-10, CIFAR-100 [24], Tiny-ImageNet [25] and ImageNet [26]. On CIFAR datasets, we employ a WideResNet-40-2 [55] (as implemented in [56]), PreAct-ResNet-18 [57] (as implemented in [58]). We use a PreAct-ResNet-18 on Tiny-ImageNet, and ResNet-18 [59] (as implemented in *torchvision* library [60]) on ImageNet.

**Implementation Details.** On CIFAR-10 and CIFAR-100, we run all experiments by five different random seeds and report the mean and standard derivation of test accuracy. We employ SGD with initial learning rate $0.1$, momentum $0.9$ and weight decay $0.0005$. We train models for 200 epochs and reduce the learning rate by $0.1$ at the 100-th and 150-th epoch with batch size of 128. We use only the standard augmentations (*i.e.*, random flipping and translation) in the basic experiments, and additionally leverage mixup [27] for further comparison. The hyperparameter $\alpha$ of mixup is set to 1 in the baseline as suggested by [27] and is properly chosen for APART. For comparison, we report the empirical results of SAM-trained WideResNet-40-2 and PreAct-ResNet-18 under standard augmentation, where we set $\rho = 0.05$ and $\rho = 0.1$ on CIFAR-10 and CIFAR-100 respectively, as suggested by [28]. On Tiny-ImageNet, we use batch size of 256 and set other hyperparameters in the same way of the CIFAR experiments; under mixup, we set $\alpha = 0.2$ for both the standard method and APART. On ImageNet, We employ SGD with initial learning rate $0.1$, momentum $0.9$ and weight decay $0.0001$. We train models with batch size of 256 for 105 epochs, where the learning rate is reduced by $0.1$ at the 30-th, 60-th, 90-th and 100-th epoch. We randomly resize and crop images to $224 \times 224$ resolution with random flipping to perform the

standard augmentation. For the hyperparameters of APART and APART-SAM, we evaluate a few combinations and show the best performance in the main results. More results of these combinations are reported in the ablation studies and appendix A.1. Considering the $2\times$ training budget of APART, we also conduct the experiments of the standard training with $2\times$ total and decay epochs to show APART's non-trivial performance. Our implementations use Pytorch [61], and all models are trained on a server with three NVIDIA RTX 3090 GPUs. Please see appendix B and the code at https://github.com/unknown9567/apart.git for more details.

## 4.2 Main Results

Table 1: Results on CIFAR-10 and CIFAR-100.

| Method (Augmentation) | Budget | CIFAR-10 | CIFAR-100 |
|---|---|---|---|
| **WideResNet-40-2** | | | |
| Standard (Standard) | $1\times$ | $94.67_{\pm 0.10}(+0.00)$ | $76.10_{\pm 0.24}(+0.00)$ |
| Standard (Standard) | $2\times$ | $94.99_{\pm 0.11}(+0.32)$ | $76.73_{\pm 0.27}(+0.63)$ |
| SAM (Standard) | $2\times$ | $95.39_{\pm 0.14}(+0.72)$ | $77.47_{\pm 0.09}(+1.37)$ |
| APART (Standard) | $2\times$ | $95.69_{\pm 0.13}(+1.02)$ | $79.05_{\pm 0.25}(+2.95)$ |
| APART-SAM (Standard) | $2\times$ | $\mathbf{95.81_{\pm 0.27}(+1.14)}$ | $\mathbf{79.21_{\pm 0.23}(+3.11)}$ |
| Standard (Mixup) | $1\times$ | $95.43_{\pm 0.11}(+0.76)$ | $76.63_{\pm 0.34}(+0.53)$ |
| Standard (Mixup) | $2\times$ | $\mathbf{96.03_{\pm 0.11}(+1.36)}$ | $77.96_{\pm 0.43}(+1.86)$ |
| APART (Mixup) | $2\times$ | $95.86_{\pm 0.05}(+1.19)$ | $\mathbf{79.22_{\pm 0.22}(+3.12)}$ |
| APART-SAM (Mixup) | $2\times$ | $95.78_{\pm 0.08}(+1.11)$ | $79.00_{\pm 0.09}(+2.90)$ |
| **PreAct-ResNet-18** | | | |
| Standard (Standard) | $1\times$ | $94.60_{\pm 0.17}(+0.00)$ | $76.30_{\pm 0.11}(+0.00)$ |
| Standard (Standard) | $2\times$ | $94.76_{\pm 0.12}(+0.16)$ | $75.34_{\pm 0.21}(-0.96)$ |
| SAM (Standard) | $2\times$ | $95.56_{\pm 0.16}(+0.96)$ | $78.57_{\pm 0.17}(+2.27)$ |
| APART (Standard) | $2\times$ | $95.84_{\pm 0.16}(+1.24)$ | $79.48_{\pm 0.15}(+3.18)$ |
| APART-SAM (Standard) | $2\times$ | $\mathbf{96.12_{\pm 0.06}(+1.52)}$ | $\mathbf{80.07_{\pm 0.18}(+3.77)}$ |
| Standard (Mixup) | $1\times$ | $95.76_{\pm 0.11}(+1.16)$ | $77.30_{\pm 0.50}(+1.00)$ |
| Standard (Mixup) | $2\times$ | $96.19_{\pm 0.12}(+1.59)$ | $78.81_{\pm 0.45}(+2.51)$ |
| APART (Mixup) | $2\times$ | $\mathbf{96.28_{\pm 0.09}(+1.68)}$ | $80.07_{\pm 0.17}(+3.77)$ |
| APART-SAM (Mixup) | $2\times$ | $96.08_{\pm 0.18}(+1.48)$ | $\mathbf{80.19_{\pm 0.15}(+3.89)}$ |

**Evaluation on CIFAR-10 and CIFAR-100.** As is shown in Table 1, APART helps models significantly outperform their counterparts trained by the standard method. Under standard augmentation, without considering the training budget, APART improves the accuracy by over $1.02\%$ on CIFAR-10 and $2.95\%$ on CIFAR-100 for each model; considering the training budget leads to the accuracy gains of over $0.70\%$ on CIFAR-10 and $2.32\%$ on CIFAR-100; enhanced by SAM, APART-SAM further improves the accuracy of APART-trained models by over $0.12\%$ on CIFAR-10 and $0.16\%$ on CIFAR-100. In addition, APART and APART-SAM outperform SAM under this experimental setting. Under mixup [27], the improvements achieved by APART are generally consistent,

Table 2: Results on Tiny-ImageNet and ImageNet.

| Method (Augmentation) | Budget | Accuracy (%) |
|---|---|---|
| **Tiny-ImageNet** | | |
| Standard (Standard) | $1\times$ | 63.52 (+0.00) |
| Standard (Mixup) | $1\times$ | 64.34 (+0.82) |
| Standard (Standard) | $2\times$ | 63.94 (+0.42) |
| Standard (Mixup) | $2\times$ | 64.54 (+1.02) |
| APART (Standard) | $2\times$ | 67.00 (+3.48) |
| APART (Mixup) | $2\times$ | 67.26 (+3.74) |
| APART-SAM (Standard) | $2\times$ | 67.53 (+4.01) |
| APART-SAM (Mixup) | $2\times$ | **68.66 (+5.14)** |
| **ImageNet** | | |
| Standard (Standard) | $1\times$ | 70.24 (+0.00) |
| Standard (Standard) | $2\times$ | 71.25 (+1.01) |
| Standard (Standard) | $4\times$ | 71.45 (+1.21) |
| APART (Standard) | $2\times$ | 70.86 (+0.62) |
| APART (Standard) | $4\times$ | **72.14 (+1.90)** |
| APART-SAM (Standard) | $2\times$ | 70.82 (+0.58) |

though the APART-trained WideResNet-40-2 is somewhat inferior to the standard counterpart with $2\times$ budget on CIFAR-10; on the other hand, APART-SAM slightly degenerates due to the potential conflicts between SAM and mixup on CIFAR datasets. Besides, mixup with sufficient training budgets boosts the standard models more significantly, reducing the accuracy gap between them and the APART-trained counterparts.

**Evaluation on Tiny-ImageNet and ImageNet.** As is shown in Table 2, APART and APART-SAM consistently improve the accuracy on ImageNet and its variant. On Tiny-ImageNet, the accuracy gains are significant, *e.g.* the models trained by APART-SAM outperforms the standard counterparts by over $4\%$ and $3.5\%$ for $1\times$ and $2\times$ training budgets respectively under standard augmentation. Furthermore, APART-SAM enjoys the combination with mixup and improves the accuracy by over $5\%$. On ImageNet, APART with $2\times$ budget outperforms the baseline with $1\times$ budget, but is inferior to the standard training with $2\times$ budget. However, scaling the training budgets leads to a different result: APART with $4\times$ budget outperforms the standard method with both $2\times$ and $4\times$ training budgets. It seems that APART employed on the large-scale dataset requires more steps to show its promise. Besides, APART-SAM slightly degenerates due to the insufficient tuning of its more hyperparameters.

### 4.3 Ablation Study

Table 3 shows the performance of the WideResNet-40-2 trained by APART with different hyperparameters in Algorithm 1 on CIFAR-100. Overall, APART-trained models outperform all standard models despite training budgets and hyperparameters. For example, even the model trained by APART with $1.19\times$ budget performs better than the standard model with $2.00\times$ budget. On the other hand, APART's hyperparameters have impacts at different levels on its performance.

**Impact of $N$.** The number of samples used in APART's second step has a significant impact on its performance. Indeed, models with APART's adversarial BN statistics implicitly generate adversarial features within the models in the second pass. Therefore, more samples in this pass lead to more diversity required by models' robustness against the noisy BN statistics and improve the performance more significantly.

Table 3: Ablation studies of APART's hyperparameters.

| Budget | $N$ | $\epsilon$ | $n$ | Accuracy (%) |
|---|---|---|---|---|
| **Standard Training** | | | | |
| $1.00\times$ | | | | $76.10_{\pm 0.24}(+0.00)$ |
| $1.20\times$ | NA | NA | NA | $76.24_{\pm 0.30}(+0.14)$ |
| $2.00\times$ | | | | $76.73_{\pm 0.27}(+0.63)$ |
| **APART** | | | | |
| | | $0.1$ | $1$ | $77.58_{\pm 0.17}(+1.48)$ |
| $1.19\times$ | $24$ | $0.1$ | $2$ | $77.50_{\pm 0.17}(+1.40)$ |
| | | $0.1$ | $8$ | $77.35_{\pm 0.36}(+1.25)$ |
| | | $0.05$ | $8$ | $78.45_{\pm 0.12}(+2.35)$ |
| | | $0.1$ | $8$ | $78.80_{\pm 0.23}(+2.70)$ |
| | | $0.2$ | $8$ | $77.95_{\pm 0.30}(+1.85)$ |
| | | $0.4$ | $8$ | $71.86_{\pm 0.12}(-4.24)$ |
| $2.00\times$ | $128$ | $0.1$ | $1$ | $78.36_{\pm 0.22}(+2.26)$ |
| | | $0.1$ | $2$ | $78.54_{\pm 0.39}(+2.44)$ |
| | | $0.1$ | $16$ | $79.05_{\pm 0.25}(+2.95)$ |
| | | $0.1$ | $32$ | $78.69_{\pm 0.19}(+2.59)$ |

**Impact of $\epsilon$.** Large perturbation radii (*e.g.* $\epsilon = 0.4$) degenerate models' performance, since strong attacks caused by such radii force the models to sacrifice their generalization for more robustness. In contrast, smaller radii reduce both the robustness and accuracy, which illustrates the link between the generalization and robustness of APART-trained models.

**Impact of $n$.** The group number has a relatively slight impact on the accuracy, since it implicitly enhances the attack. A properly chosen $n$ can help APART achieve the best performance.

### 4.4 Evaluation on APART's Attacks

**Experimental Setup.** We evaluate APART's attacks to provide a basic insight of its effectiveness. We use a WideResNet-40-2 [55] pretrained on CIFAR-100. We perform APART's first step to adversarially shift its BN statistics without changing the other parameters. We use only a batch of training samples for the attack, but evaluate the accuracy over the entire training dataset. For comparison, we provides the accuracy in the cases of random perturbations, *i.e.*, $\delta_{\mu}, \delta_{\sigma} \sim \text{Uniform}[-\epsilon, \epsilon]^d$ or

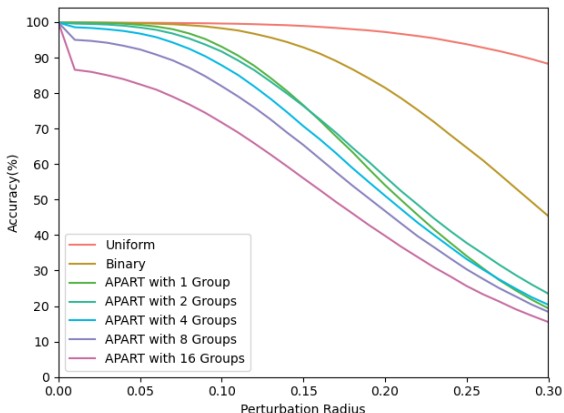

Figure 1: Evaluation on APART's Attacks.

324 randomly drawing $\delta_{\mu}, \delta_{\sigma}$ from $\{-\epsilon, \epsilon\}^d$ formed by the binary values. Besides, we test different
325 group numbers of APART to substantiate our insight of this trick.

326 **Results.** As is shown in Figure 1, the uniform random perturbations result in almost no reduction in
327 the accuracy despite the radii, while the binary random perturbations require sufficient large radii for
328 the attack. In contrast, APART uses only a batch of samples to generate the effective perturbations
329 that reduce the accuracy even under a small radius. Additionally, the larger group numbers of APART
330 provide more significant accuracy reduction especially when the radii are more limited, demonstrating
331 our insight.

### 4.5 Robustness against Perturbed BN Statistics

333 **Experimental Setup.** We evaluate the robustness of the APART-trained models against perturbed BN
334 statistics to provide the insight of APART's effectiveness. We employ the WideResNet-40-2 trained
335 by the standard method with $1\times$ and $2\times$ training budgets and APART with different perturbation
336 radii and group numbers on CIFAR-100. First, we randomly draw a direction $v$ from $\{-1, 1\}^d$ for
337 each BN statistics with the same initial random seed shared across each experiment. Second, we
338 scale $v$ by different perturbation radii $\epsilon$ to perturb the estimated BN statistics, $i.e.$, $\hat{\mu} \leftarrow (1 + \epsilon v)\hat{\mu}$ or
339 $\hat{\sigma} \leftarrow (1 + \epsilon v)\hat{\sigma}$. Then, each model with the perturbed statistics is evaluated over the test samples.

340 **Results.** As is shown in Figure 2, mod-
341 els' generalization is measured by the non-
342 perturbed accuracy, and their robustness
343 is illustrated by the accuracy reduction re-
344 sulting from the perturbations. APART-
345 trained models generally outperform the
346 standard models for both the generaliza-
347 tion and robustness. Specifically, the stan-
348 dard models (dashed lines) yield lower non-
349 perturbed accuracy and suffers from more
350 accuracy reduction as the perturbations in-
351 crease. Meanwhile, more training epochs
352 (dashed orange line) slightly improve the
353 performance of the standard methods. On
354 the other hand, APART performs better but
355 requires a trade-off between the generaliza-

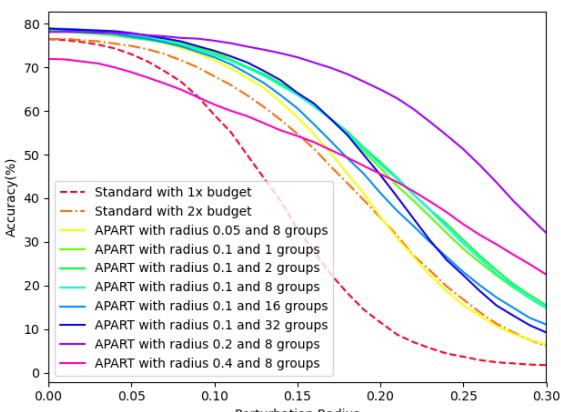

Figure 2: Robustness of WideResNet-40-2 against perturbed BN statistics on CIFAR-100.

356 tion and robustness. Increasing APART's radii improves both the robustness and generalization to
357 some extent. However, a large radius ($\epsilon = 0.4$) results in the severe degeneration (solid pink line)
358 in the generalization. Additionally, different group numbers of APART lead to improvement of
359 the generalization and robustness to varying degrees but have no clear trend. In summary, APART
360 consolidates models' robustness against noisy BN statistics to boost models' performance but requires
361 a further generalization-robustness trade-off achieved by tuning the hyperparameters.

## 5 Conclusion and Discussion

363 In this paper, we identify the robustness against the noise in BN statistics to bridge the generalization-
364 robustness gap. Then, we proposed APART that implements a new AT paradigm, termed model-
365 based AT, to achieve such robustness. APART performs attacks and defense within models by two
366 backward passes over each batch of benign samples, utilizing gradients efficiently. The empirical
367 results demonstrate APART's effectiveness in improving the robustness, which further boosts model
368 generalization on benign samples.

369 **Limitations.** Though APART improves models by solving a BN-specific problem, it and its variant
370 suffer from the potential degeneration in case of the combination with other training methods
371 implicitly involving BN, which results in more demand for fine-tuning the hyperparameters.

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
