## A  Appendix

### A.1  Complete Experimental Results

We show the complete results of APART and APART-SAM with the corresponding hyperparameters, *i.e.*, the perturbation radius $\epsilon$, group number $n$ of APART, $\rho$ of APART-SAM and $\alpha$ of mixup, in Table 4,5,6,7,8,9.

Overall, there are slight differences between the results, proving APART's insensibility to its hyperparameters. Under standard augmentation, lower perturbation radii and less group numbers lead to better performance of APART and APART-SAM on CIFAR-10; in contrast, APART and APART-SAM require larger perturbation radii and group numbers on CIFAR-100 and Tiny-ImageNet. Under mixup, APART and APART-SAM require less group numbers, which seems that there exists the potential conflict between APART's grouping trick and mixup. Besides, slightly larger $\rho$ of APART-SAM shows better performance. On ImageNet, APART and APART-SAM generally require lower values of these hyperparameters.

Table 4: Complete results of APART on CIFAR-10.

| Augmentation | $\alpha$ | $\epsilon$ | $n = 1$ | $n = 8$ | $n = 16$ |
|---|---|---|---|---|---|
| **WideResNet-40-2** | | | | | |
| Standard | NA | 0.025 | $95.52_{\pm 0.12}$ | - | $95.64_{\pm 0.10}$ |
| | | 0.05 | $95.65_{\pm 0.19}$ | - | $\mathbf{95.69_{\pm 0.13}}$ |
| Mixup | 0.2 | 0.05 | $95.73_{\pm 0.15}$ | $\mathbf{95.86_{\pm 0.05}}$ | $95.75_{\pm 0.08}$ |
| | 1.0 | 0.05 | - | - | $95.74_{\pm 0.13}$ |
| **PreAct-ResNet-18** | | | | | |
| Standard | NA | 0.025 | - | - | - |
| | | 0.05 | $95.79_{\pm 0.05}$ | $\mathbf{95.84_{\pm 0.16}}$ | - |
| Mixup | 0.2 | 0.05 | $96.22_{\pm 0.13}$ | $96.22_{\pm 0.12}$ | $96.11_{\pm 0.13}$ |
| | 1.0 | 0.05 | $\mathbf{96.28_{\pm 0.09}}$ | - | - |

Table 5: Complete results of APART-SAM on CIFAR-10.

| Augmentation | $\alpha$ | $\epsilon$ | $\rho$ | $n = 1$ | $n = 8$ | $n = 16$ |
|---|---|---|---|---|---|---|
| **WideResNet-40-2** | | | | | | |
| Standard | NA | 0.025 | 0.1 | $95.67_{\pm 0.07}$ | - | $95.77_{\pm 0.06}$ |
| | | 0.05 | 0.05 | $95.67_{\pm 0.12}$ | - | $95.66_{\pm 0.11}$ |
| | | 0.05 | 0.2 | $\mathbf{95.81_{\pm 0.27}}$ | - | $95.59_{\pm 0.08}$ |
| Mixup | 0.2 | 0.1 | 0.05 | $\mathbf{95.78_{\pm 0.08}}$ | - | - |
| | 0.2 | 0.2 | 0.05 | - | - | $95.67_{\pm 0.14}$ |
| | 0.2 | 0.2 | 0.1 | - | - | $95.70_{\pm 0.11}$ |
| **PreAct-ResNet-18** | | | | | | |
| Standard | NA | 0.05 | 0.2 | - | $\mathbf{96.12_{\pm 0.06}}$ | - |
| | | 0.05 | 0.4 | - | $96.05_{\pm 0.20}$ | - |
| Mixup | 0.2 | 0.05 | 0.1 | $\mathbf{96.08_{\pm 0.18}}$ | $95.71_{\pm 0.76}$ | - |
| | 0.2 | 0.05 | 0.2 | - | $95.87_{\pm 0.38}$ | - |

Table 6: Complete results of APART on CIFAR-100.

| Augmentation | $\alpha$ | $\epsilon$ | $n=1$ | $n=8$ | $n=16$ |
|---|---|---|---|---|---|
| **WideResNet-40-2** | | | | | |
| Standard | NA | 0.05 | - | $78.45_{\pm 0.12}$ | - |
| | | 0.1 | $78.36_{\pm 0.22}$ | $78.80_{\pm 0.23}$ | $\mathbf{79.05_{\pm 0.25}}$ |
| Mixup | 0.2 | 0.1 | $78.68_{\pm 0.19}$ | $79.00_{\pm 0.28}$ | $\mathbf{79.22_{\pm 0.22}}$ |
| | 1.0 | 0.05 | $78.08_{\pm 0.23}$ | $78.26_{\pm 0.25}$ | - |
| | 1.0 | 0.1 | - | $77.72_{\pm 0.19}$ | - |
| **PreAct-ResNet-18** | | | | | |
| Standard | NA | 0.1 | $78.94_{\pm 0.28}$ | $\mathbf{79.48_{\pm 0.15}}$ | - |
| Mixup | 0.2 | 0.1 | $\mathbf{80.07_{\pm 0.17}}$ | - | - |
| | 1.0 | 0.1 | $80.04_{\pm 0.09}$ | $79.54_{\pm 0.25}$ | - |

Table 7: Complete results of APART-SAM on CIFAR-100.

| Augmentation | $\alpha$ | $\epsilon$ | $\rho$ | $n=1$ | $n=8$ | $n=16$ |
|---|---|---|---|---|---|---|
| **WideResNet-40-2** | | | | | | |
| Standard | NA | 0.1 | 0.05 | $78.55_{\pm 0.22}$ | - | $79.16_{\pm 0.22}$ |
| | | 0.1 | 0.1 | $78.70_{\pm 0.25}$ | - | $79.19_{\pm 0.26}$ |
| | | 0.1 | 0.2 | $78.82_{\pm 0.23}$ | - | $\mathbf{79.21_{\pm 0.23}}$ |
| Mixup | 0.2 | 0.1 | 0.1 | $78.72_{\pm 0.18}$ | $78.98_{\pm 0.32}$ | $\mathbf{79.00_{\pm 0.09}}$ |
| **PreAct-ResNet-18** | | | | | | |
| Standard | NA | 0.1 | 0.2 | $79.66_{\pm 0.22}$ | $\mathbf{80.07_{\pm 0.18}}$ | $79.97_{\pm 0.46}$ |
| Mixup | 0.2 | 0.1 | 0.2 | $80.19_{\pm 0.18}$ | $\mathbf{80.19_{\pm 0.15}}$ | $79.45_{\pm 0.65}$ |

Table 8: Complete results of APART on Tiny-ImageNet and ImageNet.

| Augmentation | $\alpha$ | $\epsilon$ | $n=1$ | $n=8$ | $n=16$ |
|---|---|---|---|---|---|
| **Tiny-ImageNet** | | | | | |
| Standard | NA | 0.1 | - | **67.00** | 66.71 |
| Mixup | 0.2 | 0.05 | - | 66.74 | - |
| Mixup | 0.2 | 0.1 | 66.95 | **67.26** | - |
| **ImageNet** | | | | | |
| Standard | NA | 0.025 | **70.86** | 70.83 | - |

Table 9: Complete results of APART-SAM on Tiny-ImageNet and ImageNet

| Augmentation | $\alpha$ | $\epsilon$ | $\rho$ | $n=1$ | $n=8$ |
|---|---|---|---|---|---|
| **Tiny-ImageNet** | | | | | |
| Standard | NA | 0.1 | 0.05 | - | 67.10 |
| | | 0.1 | 0.2 | - | **67.53** |
| Mixup | 0.2 | 0.1 | 0.2 | **68.66** | 66.42 |
| **ImageNet** | | | | | |
| Standard | NA | 0.025 | 0.025 | **70.82** | - |
| | | 0.025 | 0.05 | - | 70.71 |

# B  Experimental Details

As is shown in Table 10, we summarize the hyperparameters of APART and APART-SAM used in Section 4.2, which lead to the best performance (bold numbers) of each experimental setting shown in Table 4,5,6,7,8,9.

Table 10: Hyperparameters of APART and APART-SAM for best performance in each experiment.

| Dataset | Augmentation | Model | Method | $\alpha$ | $rho$ | $\epsilon$ | $n$ |
|---|---|---|---|---|---|---|---|
| CIFAR-10 | Standard | WideResNet-40-2 | APART | NA | NA | 0.05 | 16 |
| | | | APART-SAM | | 0.2 | | 1 |
| | | PreAct-ResNet-18 | APART | | NA | | 8 |
| | | | APART-SAM | | 0.2 | | 8 |
| | Mixup | WideResNet-40-2 | APART | 0.2 | NA | 0.05 | 8 |
| | | | APART-SAM | 0.2 | 0.05 | 0.1 | 1 |
| | | PreAct-ResNet-18 | APART | 1.0 | NA | 0.05 | 1 |
| | | | APART-SAM | 0.2 | 0.1 | 0.05 | 1 |
| CIFAR-100 | Standard | WideResNet-40-2 | APART | NA | NA | 0.1 | 16 |
| | | | APART-SAM | | 0.2 | | 16 |
| | | PreAct-ResNet-18 | APART | | NA | | 8 |
| | | | APART-SAM | | 0.2 | | 8 |
| | Mixup | WideResNet-40-2 | APART | 0.2 | NA | 0.1 | 16 |
| | | | APART-SAM | | 0.1 | | 16 |
| | | PreAct-ResNet-18 | APART | | NA | | 1 |
| | | | APART-SAM | | 0.2 | | 8 |
| Tiny-ImageNet | Standard | PreAct-ResNet-18 | APART | NA | NA | 0.1 | 8 |
| | | | APART-SAM | | 0.2 | | 8 |
| | Mixup | | APART | 0.2 | NA | | 8 |
| | | | APART-SAM | | 0.2 | | 1 |
| ImageNet | Standard | ResNet-18 | APART | NA | NA | 0.025 | 1 |
| | | | APART-SAM | | 0.025 | | |