# OpenReview forum: "Adversarially Perturbed Batch Normalization: A Simple Way to Improve Image Recognition"
_NeurIPS.cc/2022/Conference — NeurIPS 2022 Submitted_

### Official Review · Reviewer_3xFg · 2022-07-08

**Rating:** 4
**Confidence:** 4
**Soundness:** 2 fair
**Presentation:** 3 good
**Contribution:** 2 fair

**Summary:**

This paper introduces an ‘Adversarially Perturbed Batch Normalization’ to improve the model’s generalization and robustness. Experiments on CIFAR, Tiny-ImageNet, and ImageNet show that the proposed methods can improve the models’ performance, compared with the baseline model.

**Questions:**

I would like to see the experimental results on the robustness benchmarks (e.g., ImageNet-C). As the main contribution lies in the model's generalization and robustness, the robustness result is must-needed, in my opinion.

**Limitations:**

For me, the current experiments are not sufficient. The authors are suggested to add more experiments to show the advantages of their paper.

**Strengths And Weaknesses:**


Strengths:
Compared with the previous AdvBN [15], the proposed APART is more appliable and easy-training.
The paper is well-written, and the theoretical analysis is clear.


Weaknesses:
Experiments
From the reviewer’s view, the experiments in this paper are not sufficient.
(1)	As mentioned in Lines 62-63, the author mentioned that they want to bridge the gap between the model’s generalization and robustness. The reviewer thinks experiments on ImageNet-C or Stylized ImageNet are needed to show the advantages of robustness.
(2)	The comparison with other methods is missing. The reviewer thinks a comparison with normalization methods [18-20] and adversarial methods [11, 15] is needed.
(3)	‘Mix-Up’ experiments on ImageNet are missing.
(4)	Similar to the experiments on CIFAR-10 and CIFRA-100, the authors are suggested to conduct the experiments on one more backbone on Tiny-ImageNet and ImageNet.

---

> ### Author Response · Authors · 2022-08-02
> **Response**
>
> We thank you for your review. Below, we address your comments. If you find our response adequate, we would appreciate it if you increase your score.
>
> * **The reviewer thinks experiments on ImageNet-C or Stylized ImageNet are needed to show the advantages of robustness.**
>
> 	We report the results of ImageNet-C as follows. For comparison, the table includes the results of AdvBN [15]. We use the [official code](https://github.com/azshue/AdvBN) to do new experiments using AdvBN, and follow the original experimental setting in its original paper [15]. Also, the table includes the results of AugMax [16] (NeurIPS’21) for further comparison. The results of APART on ImageNet-C are still significant. Note that APART does not use data augmentations. While AugMax is a composition of multiple augmentations.
>
>
>     **Table 1: Results on ImageNet and ImageNet-C.**
>
>   | **ResNet-18**                   | **ImageNet Top-1** $\uparrow$ | **ImageNet-C mCE $\downarrow$**  |
>   |---------------------------------|-------------------------------|----------------------------------|
>   | Pretrained                      | 69.76                         | 84.54                            |
>   | AdvBN [15] (Finetune 20 Epochs) | 69.81                         | 84.37                            |
>   | APART (Finetune 20 Epochs)      | 70.30                         | 84.14                            |
>   | AugMax-DuBIN [16]               | 67.62                         | 82.56                            |
>   | APART (105 Epochs)              | 70.86                         | 82.81                            |
>   | APART (210 Epochs)              | 72.14                         | 81.91                            |
>   | **ResNet-34**                   | **ImageNet Top-1** $\uparrow$ | **ImageNet-C mCE** $\downarrow$  |
>   | Standard (105 Epochs)           | 73.71                         | 76.89                            |
>   | APART (105 Epochs)              | 74.58                         | 75.17                            |
>
> * **The comparison with other methods is missing.**
>
> 	Thanks for your reminder. These normalization methods [18-20] change models' architectures indeed and are applied in the tasks with mini-batch due to the demanding computations. While APART is a training method without modifying the architectures, and focuses on the training with normal batches of samples. Lack of the comparison between them does not reduce the main contributions.
>
> 	Comparison with [15] is shown above. In terms of AdvProp [11], we will provide the comparison in the revised version. On the other hand, there is a potential combination of APART and AdvProp since they attack the different components (samples and models) in adversarial training without explicit conflicts between these two methods. Therefore, they are not competitors.
>
> * **Experiments of Mixup on ImageNet and other backbones on (Tiny-)ImageNet are missing.**
>
> 	Due to the time limit during the rebuttal, we have run some quick experiments on CIFAR datasets to evaluate APART's effects on other backbones. More experimental results on (Tiny-)ImageNet will be included in the revised version. Below is the CIFAR results on other backbones.
>
>
>     **Table 2: Results on CIFAR datasets.**
>
>   | **Model**                 | **VGG16**   | **VGG19**   | **DenseNet121**  |
>   |---------------------------|-------------|-------------|------------------|
>   | CIFAR10 (Standard/APART)  | 93.35/94.04 | 92.80/93.92 | 95.11/95.90      |
>   | CIFAR100 (Standard/APART) | 70.67/74.30 | 71.15/72.33 | 78.52/81.60      |
>
> 	As shown above, the gains on CIFAR are still significant despite the backbones. Besides, we provide the results of ResNet-34 in Table 1. The results of ResNet-34 corroborates that the deeper ResNet is still enjoying the accuracy gain brought by APART.
>
> 	We think that ResNet is a representative architecture for evaluating the performance in our paper. Based on our new experiments on CIFAR, we summarize that our proposed method indeed helps improve the generalization of the model on different backbones with mixup. We continue evaluating different backbones on ImageNet, though it is very time-consuming.

---

> > ### Comment · Reviewer_3xFg · 2022-08-09
> > **Response to the authors**
> >
> > Thanks for the rebuttal, however, the responses only address parts of my concerns. I feel grateful that the authors have added the robustness experiments on ImageNet-C, while many experiments (e.g., Mixup on ImageNet, more backbones on ImageNet) are still missing for now. In my opinion, experiments for this paper are not sufficient for publishing. I will keep my rating.

---

### Official Review · Reviewer_STBH · 2022-07-10

**Rating:** 3
**Confidence:** 4
**Soundness:** 2 fair
**Presentation:** 3 good
**Contribution:** 2 fair

**Summary:**

This paper proposes to add adversarial noise on the BN statistics to improve classification accuracy on in-distribution images.

**Questions:**

Please see above

**Limitations:**

Please see above

**Strengths And Weaknesses:**

Strength:
1. The paper is well-written and easy to follow. The related works are thoroughly discussed.

Weakness:
1. The novelty is limited. The proposed method is almost identical with AdvBN [15] (NeurIPS'21). Although the authors mentioned three differences in related work section, I still think they are all minor differences.
2. In experiments, no results on [11] or [15] are reported. This makes it hard to evaluate whether the proposed method can outperform previous works.

---

> ### Author Response · Authors · 2022-08-02
> **Response (Part1)**
>
> Thank you for your review. Below we address your comments. If you find our response adequate, we would appreciate it if you increase your score.
>
> * **The proposed method is almost identical with AdvBN [15] (NeurIPS’21).**
>
> 	We elaborate on the differences between AdvBN and APART in terms of backgrounds, motivations and implementations.
>
> 	* **Backgrounds**
>
> 		AdvBN investigates fine-tuning pretrained models to obtain the robustness against common corruptions and style changes in images. While APART aims at training models from the scratch to improve models' generalization on clean images. Also, such generalization improvement might lead to the robustness against the corruptions.
>
> 	* **Motivations**
>
> 		AdvBN attempts to generate worst-case feature perturbations during training. Then the model is trained to resist the generated feature perturbations, leading to better performance on unseen corrupted samples. While APART perturbs BN statistics from a numerical perspective instead of considering the unseen domains. Then APART helps models enhance their robustness against noise in BN statistics to obtain better generalization on clean samples. AdvBN and APART focus on completely different problems of training networks.
>
> 	* **Implementations**
>
> 		* AdvBN performs attacks in non-BN layers, e.g. the end of the 2nd convolutional stage of ResNet-50 (please refer to section 5.1 of [15]). While APART performs attacks in each BN layer. The possibly confusing similarity between them is caused by the normalization trick. Performing controllable attacks requires the normalization of internal features. So the sophisticated normalization trick of BN is adopted by AdvBN. Such normalization trick is also adopted by A-FAN [46], resulting in the similarity between AdvBN and A-FAN. While APART directly performs attacks on BN, which results in the confusing similarity between AdvBN and APART's perturbation formulas.
>
> 		* As described in Algorithm 1 of [15],  AdvBN freezes the shallower subnetwork $g_{\theta}^{1,l}$ to extract the stable features, and only fine-tunes the deeper subnetwork $g_{\theta}^{l+1,L}$ . Thus, the feature perturbations of AdvBN essentially share the same paradigm with sample perturbations, where the major difference is that AdvBN attacks features (as special samples fed into internal layers) instead of samples fed into input layers. APART follows a different paradigm that attacks the model instead of samples. In detail, APART's attacks are performed in all BN layers through the whole network instead of the features in a specified single layer. Besides, APART is not a multi-layer version of AdvBN. The authors of AdvBN tried multiple AdvBN layers, but found that the perturbations at successive layers compound and destabilize the training process (please refer to "Reply to Reviewer 4gjW" of AdvBN's [OpenReview](https://openreview.net/forum?id=A-RON3lv-aR)). In contrast, APART is stable under its multi-layer attacks.
>
> 		* As discussed before, AdvBN aims to fine-tune pretrained models to obtain the robustness. On the other hand, employing AdvBN to train models from the scratch will contradict with its assumptions, since AdvBN performs attacks on the semantic features extracted by some pretrained models. Thus, AdvBN heavily depends on pretrained models. While APART is proposed to train models from the scratch, and the pretrained models can also be used in the training. For the next question about the experiments, we will report the results of comparing these two methods.

---

> ### Author Response · Authors · 2022-08-02
> **Response (Part2)**
>
> * **No results on [11] or [15] are reported.**
>
> 	Due to the time limit during the rebuttal, we compare with AdvBN [15] currently. For AdvProp [11], we will provide the comparison in the revised version. On the other hand, there is a potential combination of APART and AdvProp since they attack the different components (samples and models) in adversarial training without explicit conflicts between these two methods. Therefore, they are not competitors.
>
> 	We use the [official code](https://github.com/azshue/AdvBN) to do new experiments using AdvBN, and follow the original experimental setting in its original paper [15].  Besides, we fine-tune pretrained ResNet-18 by APART with the same epochs and learning rate. The performance improvements brought by APART are significant in the results shown as follows (including APART's results of training from the scratch).
>
>     **Table 1:  Results on ImageNet and ImageNet-C.**
>
>   | **ResNet-18**                   | **ImageNet Top-1** $\uparrow$ | **ImageNet-C mCE $\downarrow$**  |
>   |---------------------------------|-------------------------------|----------------------------------|
>   | Pretrained                      | 69.76                         | 84.54                            |
>   | AdvBN [15] (Finetune 20 Epochs) | 69.81                         | 84.37                            |
>   | APART (Finetune 20 Epochs)      | 70.30                         | 84.14                            |
>   | AugMax-DuBIN [16]               | 67.62                         | 82.56                            |
>   | APART (105 Epochs)              | 70.86                         | 82.81                            |
>   | APART (210 Epochs)              | 72.14                         | 81.91                            |
>   | **ResNet-34**                   | **ImageNet Top-1** $\uparrow$ | **ImageNet-C mCE** $\downarrow$  |
>   | Standard (105 Epochs)           | 73.71                         | 76.89                            |
>   | APART (105 Epochs)              | 74.58                         | 75.17                            |

---

### Official Review · Reviewer_fuBX · 2022-07-11

**Rating:** 5
**Confidence:** 4
**Soundness:** 3 good
**Presentation:** 4 excellent
**Contribution:** 3 good

**Summary:**

While Adversarial Training is one of the most successful methods to increase robustness, it usually degrades performance of the models on clean images. The authors attribute this to distributional discrepancy in Batch Norm statistics. They propose Adversarially Perturbed bAtch noRmalizaTion (APART) to achieve robustness against BN statistics noise, and to bridge the gap between models’ generalization and robustness. They perform backward passes twice over each batch of clean samples. The first backward pass produces two gradient computations: a normal gradient that helps update parameters of model, and a statistics gradient that is used to perturb the statistics parameters in BN. The second pass is performed to generate the defensive gradient that helps the model resist the adversarial statistics perturbation. The normal and defensive gradients are combined to improve both generalization and robustness of the model. Experiments are performed on CIFAR, Tiny-ImageNet and ImageNet, and show improved clean accuracy over standard training and SAM [28].




**Questions:**

See the Clarity section above.

**Limitations:**

- The authors have addressed limitation of the work in terms of suffering from potential degeneration in case of the combination with other training methods implicitly involving BN.
- The authors can address potential limitation of their work on large-scale datasets and models.
- There are no potential negative societal impact that need to be specifically addressed.

**Strengths And Weaknesses:**

**Originality and Significance**:
- The paper presents a new way of bridging the gap between models’ generalization and robustness. It is known in the literature that there is discrepancy between Batch Norm statistics of clean and adversarial examples [13] (as well as the statistics from different batches). AdvProp proposes using two batch norm statistics, one for clean images and one auxiliary for adversarial examples [13]. Rather than creating a separate layer to deal with this discrepancy, the paper attempts to make the models robust to the BN statistics noise. This approach is interesting and novel to the best of my knowledge.
- The method can be combined with other augmentations to further boost performance. The proposed combination with SAM (as one of the state-of-the-art methods) is particularly promising.

**Quality**:
- Overall the paper is well-structured and well-written.
- The proposed approach is sound, and is described clearly.
- Experiments are performed on various datasets including CIFAR-10, CIFAR-100, Tiny-ImageNet and ImageNet. Overall, experimental results are convincing. They demonstrate improvements on clean accuracy over the baselines as well as robustness against perturbed BN statistics. Comparing with baselines using the same budget is important given the additional cost of the proposed approach.
- The authors report detailed experimental results in the supplementary material, and show that ARAPT is relatively insensitive to hyper-parameters.

**Clarity**:
- Scalability of ARAPT to large datasets and models is not clearly supported in the experiments. The authors use the relatively small ResNet-18 model on ImageNet-1K. ARAPT underperforms standard training on ImageNet at 2x budget and outperforms it at 4x budget (Table 2). The authors note that "APART employed on the large-scale dataset requires more steps to show its promise", but do not provide further explanation or experiments on this.
- All experiments are performed on the ResNet family. On ImageNet the achieved accuracy of 72.14% (Table 2) is far from the state-of-the-art. It'd be good to include experiments on other architectures (e.g. EfficientNet), and see if the gains are significant.

---

> ### Author Response · Authors · 2022-08-02
> **Response**
>
> Thank you for your review. We are glad that you liked our paper. Below, we address your concerns with the paper. If you find our response adequate, we would appreciate it if you increase your score.
>
> * **Scalability of ARAPT to large datasets and models is not clearly supported in the experiments.**
>
> 	As reported in the paper, with the same 2x budget, APART is inferior to the standard training on ImageNet: APART improves the accuracy by 0.62% while scaling the epochs of standard training leads to the improvement of 1.01%. However, such significant improvement (over 1%) from scaling the epochs illustrates the underfitting of the model in this experiment. Indeed, APART focuses on improving generalization of models trained with sufficient epochs, and to some extent raises the accuracy of insufficiently trained models. The reported further experiments with 4x budget substantiate the effects of APART: standard method merely leads to 1.21% while APART leads to 1.90% in accuracy improvements.
>
> 	Such phenomenon also exists in SAM [28]. As reported in Table 2 therein [28], in terms of shallower ResNet-50, SAM with 100 epochs is inferior to the standard training with 200 epochs though they share the same training budget. This phenomenon disappears in deeper networks since they fit the samples more easily.
>
> 	Intuitively, in the AT paradigm followed by APART and SAM, some gradients are used to perform attacks that inevitably slow the convergence of training. Therefore, sufficient epochs are needed.
>
> 	Besides, there is a link between networks' depths and BN's effects on training. Such link leads to varying effects of APART on different backbones. We conduct the experiment of ResNet-34 after submission, where APART with 2x budget improves the accuracy by 0.87%, more significant than 0.62% in terms of ResNet-18. We will include more experimental results in the revised version.
>
> * **It'd be good to include experiments on other architectures (e.g. EfficientNet), and see if the gains are significant.**
>
> 	Due to the time limit during the rebuttal, we have run some quick experiments on CIFAR datasets to evaluate APART's effects on other architectures. More results on EfficientNet and ImageNet will be included in the revised version. The gains on CIFAR are still significant despite the architectures. Below is the CIFAR results.
>
>     **Table 2:  Results on CIFAR datasets.**
>
>   | **Model**                 | **VGG16**   | **VGG19**   | **DenseNet121**  |
>   |---------------------------|-------------|-------------|------------------|
>   | CIFAR10 (Standard/APART)  | 93.35/94.04 | 92.80/93.92 | 95.11/95.90      |
>   | CIFAR100 (Standard/APART) | 70.67/74.30 | 71.15/72.33 | 78.52/81.60      |

---

### Official Review · Reviewer_sCRg · 2022-07-11

**Rating:** 4
**Confidence:** 3
**Soundness:** 2 fair
**Presentation:** 3 good
**Contribution:** 3 good

**Summary:**

The paper proposes a method to improve the generalization of neural networks by training them to be robust to adversarial perturbations in the statistics of the batch normalization (BN) layers. The approach combines gradients computed on unperturbed BN statistics with gradients computed on perturbed statistics. Perturbations or noise in the BN statistics are obtained through 1) signed gradients from the first update and 2) reductions in the batch size for the second update.
Experiments demonstrate improvements over standard training, especially in the case of smaller-scale datasets, i.e., CIFAR and Time-ImageNet. The method can also be combined with other techniques, such as Mixup and SAM optimization, typically leading to further improvements.

**Questions:**

I would appreciate it if the authors could address the weakness listed above, especially the first three points, i.e.,
- How is the robustness to other typical adversarial attacks?
- How is the introduced AT paradigm different from what prior works proposed, or what novel insights are provided?
- What are the issues with large-scale training?

**Limitations:**

As mentioned above, it might be good to further address the performance on larger scale datasets if this turns out to be a limitation. Also, depending on how robust the method is to other adversarial perturbations, this could also be mentioned in the limitations.

**Strengths And Weaknesses:**

Strengths:
- The method benefits the generalization of neural networks trained on smaller datasets considerably
- The technical presentation of the method in Section 3.2 is detailed and sufficiently clear
- The method can be combined with other training methods, such as SAM.


Weaknesses:
- The paper claims to bridge the gap between robustness and generalization. Experiments are focused mainly on the generalization ability of the learned networks, and robustness experiments are restricted to perturbations of the BN statistics. This is quite limited, and it is unclear if the learned networks are robust to various other adversarial attacks. Indeed, it is unclear what the relevance of Sections 4.4 and 4.5 are regarding the robustness of the networks in practice.
- Another contribution of the paper is "a new AT paradigm, termed model-based AT." It appears that the main idea of perturbing model parameters has been explored in various prior works (e.g., [8, 28]). It is not clear what the generic formulation in Eq 2 contributes or what novel insights are provided.
- The benefits of the method seem to disappear during large-scale experiments on ImageNet. This is somewhat concerning, and it might be good to investigate this issue further.
- Section 3.3 is somewhat confusing: L206 claims \mathcal{R}=0, but then \mathcal{R} appears in the perturbation computation of (7). It is also unclear if a term similar to g_\phi exists in this case.

---

> ### Author Response · Authors · 2022-08-02
> **Response (Part1)**
>
> Thank you for your detailed comments. As prior works [11, 15] did, this work focuses on improving models' performance in standard classification on benign samples, without considering their robustness against adversarial samples. Below, we address your only concerns with the paper. If you find our response adequate, we would appreciate it if you increase your score.
>
> * **How is the robustness to other typical adversarial attacks?**
>
>     We suspect that the reviewers may have misunderstood our main concern. Indeed, robustness against adversarial attacks is an important problem in machine learning. But, it mainly belongs to the noise perturbation on the input image. In this paper, we mainly study the robustness of the network architecture, especially for the BN layer, which affects the generalization of the test data.
>
>     Generally, the definition of robustness varies in different contexts: for tasks with safety concerns, it's defined as the robustness against adversarial samples; for standard classification without safety concerns, the definition is unclear without explicit assumptions. Some prior works [15,16,17] reposition the robustness as the one against common corruptions. Besides, robustness might be models' stability for some problems, or models' insensitivity to some noise.
>
>     For standard classification, we focus on BN layers and identify the robustness as the one against noise in BN statistics in the submitted paper. Then we concentrate on such robustness in attempts to improve models' generalization.
>
>     Empirically, the results of section 4.4 and 4.5 corroborate our method's effects of achieving the robustness against BN statistics noise. Meanwhile, our method can boost models' generalization as shown in section 4.2. These experimental results prove our method has bridged the robustness-generalization gap.
>
>     Moreover, such robustness obtained by the proposed method has a positive impact on the models' performance on corrupted samples (considered as corrupted attacks). During the rebuttal, we have run quick experiments of the evaluation on ImageNet-C, with the results shown as follows.
>
>
>     **Table 1: Results on ImageNet and ImageNet-C.**
>
>     | **ResNet-18**                   | **ImageNet Top-1** $\uparrow$ | **ImageNet-C mCE $\downarrow$**  |
>     |---------------------------------|-------------------------------|----------------------------------|
>     | Pretrained                      | 69.76                         | 84.54                            |
>     | AdvBN [15] (Finetune 20 Epochs) | 69.81                         | 84.37                            |
>     | APART (Finetune 20 Epochs)      | 70.30                         | 84.14                            |
>     | AugMax-DuBIN [16]               | 67.62                         | 82.56                            |
>     | APART (105 Epochs)              | 70.86                         | 82.81                            |
>     | APART (210 Epochs)              | 72.14                         | 81.91                            |
>     | **ResNet-34**                   | **ImageNet Top-1** $\uparrow$ | **ImageNet-C mCE** $\downarrow$  |
>     | Standard (105 Epochs)           | 73.71                         | 76.89                            |
>     | APART (105 Epochs)              | 74.58                         | 75.17                            |

---

> ### Author Response · Authors · 2022-08-02
> **Response (Part2)**
>
> * **How is the introduced AT paradigm different from what prior works proposed, or what novel insights are provided?**
>
>     The prior works [8, 28] focus on the trainable parameters of a model, i.e., the weights, and optimize them to find a minima with a flat loss landscape. In their methods, only the attacks and defense over the trainable parameters are considered. However, there exist non-trainable parameters in models, e.g. BN statistics (mean and variance), which require estimation instead of optimization. So we proposed the new AT paradigm defined by Eq 2. This paradigm allows the attacks and defense on both trainable and non-trainable parameters, and further enables the combination of APART and SAM [28].
>
> * **What are the issues with large-scale training?**
>
>     As reported in the paper, with the same 2x budget, APART is inferior to the standard training on ImageNet: APART improves the accuracy by 0.62\% while scaling the epochs of standard training leads to the improvement of 1.01%. However, such significant improvement (over 1%) from scaling the epochs illustrates the underfitting of the model in this experiment. Indeed, APART focuses on improving generalization of models trained with sufficient epochs, and to some extent raises the accuracy of insufficiently trained models. The reported further experiments with 4x budget substantiate the effects of APART: standard method merely leads to 1.21% while APART leads to 1.90% in accuracy improvements.
>
>     Such phenomenon also exists in SAM [28]. As reported in Table 2 therein [28], in terms of shallower ResNet-50, SAM with 100 epochs is inferior to the standard training with 200 epochs though they share the same training budget. This phenomenon disappears in deeper networks since they fit the samples more easily.
>
>     Intuitively, in the AT paradigm followed by APART and SAM, some gradients are used to perform attacks that inevitably slow the convergence of training. Therefore, sufficient epochs are needed.
>
>     Besides, there is a link between networks' depths and BN's effects on training. Such link leads to varying effects of APART on different backbones. We conduct the experiment of ResNet-34 after submission, where APART with 2x budget improves the accuracy by 0.87%, more significant than 0.62% in terms of ResNet-18. We will include more experimental results in the revised version.
>
> * **Section 3.3 is somewhat confusing.**
>
>     Sorry for the confusing formula. $\mathcal{R}$ in L206 differs from the latter $\mathcal{R}$ in Eq 7. The former is introduced to formulate the prior method [28] in our proposed AT paradigm, and the latter is exactly  $\mathcal{L}(x, y; \theta, \phi)$ with $\theta = 0$ defined by Eq 4 in section 3.2. Generally, there are minor differences between APART and SAM [28] in terms of their objective functions $\mathcal{R},\mathcal{L}$. In the revised version, we will replace $R$ in L206 with $R_{sam}$ to make it clear. Meanwhile, we will replace $\mathcal{L}$ in R205 with $\mathcal{L}_{sam}$.

---

### Author Response · Authors · 2022-08-02
**General Response**

We thank all reviewers for their critical assessment of our work. This work focuses on improving models' performance in standard classification on benign samples, without considering their robustness against adversarial samples.  To this end, we would like to discuss the definition of robustness that is somewhat confusing in the submitted version, and then highlight our contributions.

Indeed, the definition of robustness varies in different contexts: for tasks with safety concerns, it's defined as the robustness against adversarial samples; for standard classification without safety concerns, the definition is unclear without explicit assumptions. Some prior works [15,16,17] reposition the robustness as the one against different common corruptions. Besides, robustness might be the models' stability for some problems or models' insensitivity to some noise.  For example, the noise in the classical Batch Normalization (BN) layer affects the performance generalization of the test data. We find that few works aim to solve this problem, so this paper aims to handle this problem.

To this end, we highlight our contributions as follows:

1. We identify the robustness against the noise in BN statistics. Enhancing such robustness by adversarial training (AT) improves the generalization on benign samples. We bridge the gap between models' generalization and the robustness.
2. We proposed APART following an AT paradigm to achieve such robustness. Empirically, models trained by APART are robust to BN statistics noise and meanwhile they enjoy significant accuracy gains. These empirical results substantiate our insights of the identified robustness.
3. APART has plug-and-play nature that allows the combination with other training methods, which leads to further accuracy gains.

---

### Meta-Review · Area_Chair_TkCE · 2022-08-28

**Recommendation:** Reject
**Confidence:** Certain

**Metareview:**

The paper presents a new way of bridging the gap between models’ generalization and robustness, by combining gradients computed on unperturbed BN statistics with gradients computed on perturbed statistics. The main goal is to improve the standard generalization, but the authors should clarify their definition of "robustness" as it seems to confuse all reviewers (e.g., questioning adversarial attacks). Moreover, the method itself is very simple, and the idea of using adversarial perturbation to stabilize model training isn't new (AdvProp, etc.). Reviewers are further concerned about the lack of large-scale experiments or on state-of-the-art architectures. Besides, there are no comparisons with some of the competing methods such as AdvProp. Therefore, I find no sufficient ground to recommend acceptance in this paper's current shape.

**Award:**

No

---

### Decision · Program_Chairs · 2022-09-14

Reject